**Data Availability Statement:** Data for this study are only available upon request. There are ethical restrictions on sharing our de-identified data set publicly. Participants in our study did not consent

# "There is a strangeness in this disease": A qualitative study of parents' experiences caring for a child diagnosed with COVID-19

Samantha Louie-Poon[1], Kathy Reid[1], Priscilla O. Appiah[1], Lisa Hartling[2], Shannon D. Scott[1]*

1 Faculty of Nursing, University of Alberta, Edmonton, Alberta, Canada, 2 Department of Pediatrics, Faculty of Medicine & Dentistry, University of Alberta, Edmonton, Alberta, Canada

* shannon.scott@ualberta.ca

## Abstract

### Background

The beginning of the COVID-19 pandemic marked a period of uncertainty as public health guidelines, diagnostic criteria, and testing protocols or procedures have continuously evolved. Despite the virus being declared a worldwide pandemic, little research has been done to understand how parents manage caring for their child diagnosed with COVID-19. We sought to understand parents' experiences and information need when caring for a child diagnosed with COVID-19.

### Methods

A qualitative descriptive study with an inductive and exploratory approach was completed. Participants were recruited through social media and local public health clinics. Data collection and analysis were concurrent. Semi-structured virtual interviews were conducted with 27 participants. Thematic analysis was conducted.

### Findings

Four major themes emerged: a) dealing with uncertainty; b) anxiety; c) social stigma and stress; d) a sense of community.

### Conclusion

Our study highlights that parent experiences were diverse and multi-faceted, and their experiences evolved and shifted over the course of the pandemic. Parents would benefit from clear and consistent evidence-based online information. Understanding the perspectives of parents caring for a child with COVID-19 is an important step in developing future resources tailored to meet their unique experiences and information needs.

to share their data on repositories, and therefore, the ethics committee have imposed these restrictions. This study has been reviewed by a Research Ethics Board 1 at the University of Alberta (#Pro0062904). Should you need to contact the Research Ethics Office, their contact information is: reoffice@ualberta.ca.

**Funding:** This study was funded by the Canadian Institutes of Health Research grant number GA3177729 (SDS, LH). https://cihr-irsc.gc.ca/e/193.html. The funders had no role in study design, data collection and analysis, decision to publish, or preparation of the manuscript.

**Competing interests:** The authors have declared that no competing interests exist.

## Introduction

In late November, 2019 a new strain of coronavirus was identified. This virus quickly spread and by March 2020, COVID-19 was declared a worldwide pandemic [1]. Little was known about the virus at that time, including how it was spread, and who would be most susceptible to severe infections [2]. Many infectious disease specialists and pediatricians agreed that, in the early phases of the pandemic, children were not becoming as severely ill when infected as adults [3–11]; however, many public health decisions, such as closing of all in-person learning, directly impacted children [12, 13], including the potential for worsening existing mental health problems [14]. In Canada, all schools were closed to in-person learning at some point during the pandemic. This affected virtually all aspects of child development, resulting in deteriorations in mental health due to not only loss of learning but also isolation from peers, teachers, coaches, and extended family [15]. As the virus evolved, new variants came in waves. Each variant raised questions about who had the potential to become seriously ill, and very little was known throughout the waves on how ill children could become from this novel virus [16].

In December 2020, Canada began administering COVID-19 vaccine to adults–starting with the most elderly Canadians. Vaccinations of adult populations continued with children being the last groups to be eligible for vaccination. Vaccination of adolescents between the ages of 12 to 17 years began in May 2021, with children ages 5 to 12 years beginning in November 2021. Vaccination of children between the ages of 6 months and 5 years only began in August 2022.

Children, since the pandemic began, continue to be diagnosed with COVID. Diagnosis is based primarily on symptoms, which have evolved over time [17]. Obtaining testing for children to confirm the diagnosis has been a challenge. The "gold standard" test is the PCR (polymer chain reaction)—a test which involves obtaining a nasopharyngeal aspirate [18, 19]. This invasive test can be difficult to obtain on young children, and many testing sites did not have pediatric expertise making it challenging for parents to have the diagnosis confirmed.

Given the ever-evolving nature of this novel disease, the changing symptoms and diagnostic uncertainty, there is a rapidly evolving body of evidence on how the different variants may affect the health of children, and how the disease may be managed. Parents have had to live with this uncertainty while managing their children during lock-downs, virtual learning, and other public health measures. Little research has been done to understand how parents managed caring for their children during this time. We sought to understand parents' experiences and information needs of caring for a child diagnosed with COVID-19. Understanding their experiences is an important step in developing future resources tailored to meet their information and support needs.

## Methods

### Study design

This was a qualitative descriptive study with an inductive and exploratory approach [20].

### Sample

Starting in spring 2020 participants were recruited through social media accounts (Twitter, Facebook, and Instagram) of the research program and through posters that were placed at local public health clinics where children and families were being tested for COVID-19. Participants were eligible if they were the parent/guardian of a child under the age of 18 who was diagnosed with COVID-19. Eligibility criteria included the ability to speak and read English, and access to Zoom. All interested parents/guardians emailed the research staff who completed screening (S1 File) via email. Eligible participants were then invited to be interviewed.

## Data collection

Data collection and analysis occurred concurrently [21]. Researchers trained in qualitative data collection contacted participants via email to set up interviews at a mutually agreed upon date and time. Given that in-person interviews were not possible during the pandemic all interviews were conducted using the Zoom videoconferencing platform. Zoom has been shown to be a viable tool as it is relatively easy to use, cost effective, and has both data management features and security options [22]. Interviews were conducted by two members of the research team (SDS, KR) that are healthcare professionals and trained in qualitative research. Caregivers provided electronic written signed consent, and when the interview was confirmed, they completed demographic information electronically (S2 File). Demographic information collected included the number of children in the family, however in the interviews we focused on the children who had COVID. A semi-structured interview guide was used to capture their experiences of caring for a child with COVID-19 (S3 File). Interviews began with the interviewer explaining their professional qualifications as Registered Nurses (RNs) (SLP, KR, PA, SS), their experience in conducting research (Graduate Students: SLP, PA; Project Coordinator SLP, KR; Principal Investigators LH, PhD and SS, PhD) and their reasons for doing this study. Interview questions moved from general to more specific with later interviews becoming more focused. In addition to recording, the interviewers completed field notes during the interviews. As data collection and analysis occurred concurrently, data collection continued until no new themes were identified, that is data redundancy was achieved [23]. Notably, at around the same time children became eligible for vaccination and the COVID-19 pandemic had evolved to become more endemic.

## Data analysis

All interviews were digitally recorded, professionally transcribed verbatim, and cleaned for accuracy and completeness. Interviewer field notes were reviewed to ensure accuracy of the recordings. Cleaned transcripts were uploaded to NVivo data management system. Inductive thematic data analysis [24] was performed by two research team members (SLP, POA) trained in qualitative data analysis methods following a multi-stage process. First, all transcripts were read and re-read in detail (SLP, POA). Second, SLP and POA conducted free coding of the first four transcripts and grouped codes into preliminary categories. Research team members (SLP, POA, SDS) reviewed and discussed codes and preliminary categories. Third, preliminary categories were grouped into a coding framework from recurring themes that emerged from the interviews. Fourth, all transcripts were coded according to the coding framework developed. Research team members (SLP, POA, SDS) met every two weeks to discuss the analysis process and reviewed themes. The coding framework was iteratively updated and modified as new themes emerged from the transcripts. Preliminary findings and interpretations were continuously reviewed and discussed among the research team.

Trustworthiness was maintained using multiple techniques to ensure: credibility, dependability, transferability, and confirmability [25]. Credibility was maintained by recruiting a diverse sample, and engaging in investigator triangulation. All team members documented all analytical decisions, modifications, and impressions in a comprehensive study log throughout the study to ensure dependability. Transferability is facilitated by providing rich description of the research participants and research findings, including direct quotes from participants to support the themes. Investigator self-reflexivity was utilized to ensure confirmability.

## Ethics

Ethics approval was obtained from the University of Alberta Health Research Ethics Board (#Pro0062904) prior to starting recruitment.

## Results

### Demographic characteristics

Thirty-one parents who met eligibility criteria agreed to participate. Twenty-seven virtual individual interviews were conducted with parents between May 2020 and April 2022. Four parents did not attend their scheduled interview and did not respond to follow up email. The interviews ranged in length from 13 minutes to 48 minutes, with a mean time of 27 minutes. Participants' demographic characteristics are presented in Table 1.

### Themes

Four major themes emerged from our analysis (Table 2): a) dealing with uncertainty; b) anxiety; c) social stigma and stress; d) a sense of community.

Each theme, subtheme, and supporting quotes are discussed in more detail below. A thematic analysis sample for theme 1 is available in Table 3.

**Theme 1: Dealing with uncertainty.** Parents of children diagnosed with COVID-19 described experiences navigating the uncertainty of the pandemic given the novelty of the virus. This uncertainty included rules and recommendations that were regularly changing, and symptoms of COVID-19 that changed or were different for children and adults. Parents expressed concern over trust when dealing with different sources of information–including health care providers who were also learning about COVID-19 at the same time. This theme consisted of the following subthemes: public health measures and symptoms of COVID-19.

*Subtheme 1*: *Public health measures.* The majority of the parents in our study expressed uncertainty with COVID-19 public health measures, including isolation rules and testing procedures. It was common for parents to search for COVID-19 public health guidelines, such as isolation measures, by "mostly piecing it together from what [they] saw online" (Interview 18, parent of a 23- month-old). Parents expressed that uncertainties arose due to COVID-19 guidelines that were not always consistent:

> "*it's just so much information and it doesn't seem to be continuous. Like one person will have one aspect–like one take–on it, and another person will have another take on [COVID-19 isolation guidelines] even in the medical community, and that I find hard. I think that's what we found hard.*" (Interview 10, parent of a 13-year-old)

Additionally, parents often expressed that the public health measures regarding isolation rules and testing procedures did not evolve as the pandemic evolved. One parent said "if I had actually followed the guidelines, I never would've taken him (the six-year-old) to get tested" (Interview 16, parent of a one-month-old, a four-year-old and a six-year-old). For some parents, the lack of consistent and evolving messages was also heightened by unclear information as, demonstrated by the following comment:

> "*most confusing part at that time was what to do with our close contact daughter, who had no symptoms, was not positive, had not been around her brother for his infectious period.*" (Interview 25, parent of an eight-year-old)

*Subtheme 2*: *Symptoms of COVID-19.* Challenges with navigating the unknowns were not restricted to the public health measures. Parents expressed uncertainty when navigating the physical signs and symptoms of COVID-19. As one participant stated, "there is a strangeness in this disease" (Interview 15, parent of a three-year-old and a six-year-old). Parents often found it stressful to navigate the physical symptoms of COVID-19 because they were

**Table 1. Demographic characteristics of sample of parents whose child had COVID-19 (n = 25[a]).**

| Characteristic | n (%) |
|---|---|
| Sex | |
| Female | 22 (85%) |
| Male | 4 (15%) |
| What is your relationship to the child? | |
| Parent | 26 (100%) |
| Grandparent | 0 (0%) |
| Guardian | 0 (0%) |
| Other | 0 (0%) |
| Age | |
| Less than 20 years | 0 (0%) |
| 21–30 years | 3 (12%) |
| 31–40 years | 14 (54%) |
| 41–50 years | 8 (31%) |
| Over 50 Years | 1 (4%) |
| Marital Status | |
| Married/Partnered | 25 (96%) |
| Single/Separated/Divorced/Widowed | 1 (4%) |
| Education | |
| High school diploma | 1 (4%) |
| Post-secondary certificate/diploma | 5 (20%) |
| Post-secondary degree | 9 (35%) |
| Graduate degree | 11 (41%) |
| Household Income | |
| Less than $25,000 | 1(4%) |
| $25,000-$49,000 | 0 (0%) |
| $50,000-$74,000 | 0(0%) |
| $75,000-$99,000 | 6 (23%) |
| $100,000-$149,000 | 3 (12%) |
| $150,000 and over | 12 (46%) |
| Prefer not to answer | 4 (15%) |
| Number of Children in the Family | |
| 1 | 6 (23%) |
| 2 | 11 (42%) |
| 3 | 6 (23%) |
| 4 | 3 (12%) |
| Does the child have other medical conditions? | |
| Yes | 3(12%) |
| No | 23 (88%) |
| Did the child require ED Care? | |
| Yes | 5 (19%) |
| No | 20 (77%) |
| Missing | 1 (4%) |
| Was the child admitted to the hospital? | |
| Yes | |
| No | |
| Missing | |

[a] A total of 27 participants were included in this study; however, only 25 participants submitted their demographic information.

**Table 2. Qualitative themes.**

| Theme | Subthemes |
|---|---|
| Dealing with uncertainty | Public health measures |
|  | Symptoms of COVID-19 |
| Anxiety |  |
| Social stigma and stress | Stigma changing over time |
|  | Social factors of stress |
| A sense of community |  |

"constantly waiting for [their child] to get worse" (Interview 17, parent of an 18-month-old). This was compounded by the fact that parents felt their "mind kind of goes to the worst-case scenario" (Interview 21, parent of a four-year-old). Parents also discussed the uncertainty if their child had symptoms that did not match the information they could find, and how the information provided was at times poorly written and difficult to read.

For some parents, healthcare providers were a positive source of information. One parent explained that "our doctor. . .was really good. She called us, like her next day that she was in, the Monday. And she called us and told us about [symptoms] . . . and what to watch for" (Interview 12, parent of a six-year-old). While other parents expressed that their doctors could provide better support during the uncertainties:

> "I think there was, you know, a line in one of those information pages where it was like, you know, notify your doctor. But I felt like that would have been really useful if my doctor would check in everyday. . .just saying 'this is what to expect, this is when to call'" (Interview 5, parent of a 14-year-old).

Several parents in our study discussed how they wished their health care providers would have reached out to them, especially in early waves when COVID testing was still available.

**Table 3. Thematic analysis example.**

| Code and Definition | Subtheme | Theme |
|---|---|---|
| Limited or conflicting COVID-19 information: lack of trusted or evidence-based sources of information, lack of easy to follow and step by step information and/or instructions for the general public. | Public Health Measures | Dealing with Uncertainty |
| Supporting Quotes: |  |  |
| "it's just so much information and it doesn't seem to be continuous. Like one person will have one aspect–like one take–on it, and another person will have another take on [COVID-19 isolation guidelines] even in the medical community, and that I find hard. I think that's what we found hard." (Interview 10, parent of a 13-year-old) |  |  |
| "most confusing part at that time was what to do with our close contact daughter, who had no symptoms, was not positive, had not been around her brother for his infectious period." (Interview 25, parent of an eight-year-old) |  |  |
| General uncertainty about COVID-19 symptoms or measures to take, parents lacking knowledge on COVID-19 or uncertain about recurrence of COVID, general unknowns about future or progression of COVID. | Symptoms of COVID-19 |  |
| Supporting Quotes: |  |  |
| "I'm not sure that this is COVID. Maybe she's got another infection". (Interview 26, parent of a 16-month-old and a six-year-old) "constantly waiting for [their child] to get worse" (Interview 17, parent of an 18-month-old). |  |  |

Parents suggested that having more frequent check ins from health care providers would have been helpful due to the newness of this illness. Over time, as testing was no longer available, parents were relying on home tests to determine if their child had COVID, leading to increased uncertainty. One parent (Interview 26, parent of a 16-month-old and a six-year-old) reported that the doctor said (about the 16-month-old) "I'm not sure that this is COVID. Maybe she's got another infection".

**Theme 2: Anxiety.** The theme of anxiety related to the expression of both parents and children. Parents felt anxious related to the diagnosis and worried about how sick their child could become over time. Parents also reported that their children expressed fear of the disease as well as worry about spreading the disease to others.

Parents expressed that their anxiety decreased following the COVID-19 diagnosis due to the asymptomatic or mild nature of the symptoms their child presented with. Twenty-four of the 27 parents in our study reported asymptomatic or mild symptoms with minimal disruption to their child's daily activities. For some, their child did not present with specific symptoms and "he maybe had one day. . .a bit of a cough and tired" (Interview 24, parent of an 11-year-old and a 13-year-old). Other parents noticed their child presenting with mild symptoms over a short duration: "she was pretty fussy. She had a little bit of a runny nose so I think that was the symptom she had, was that really. I did one day take her temperature because she was fussy, and it was like 38, so it was a little high, but it was for one day" (Interview 11, parent of a two-year-old). In some cases, parents reported their child having multiple mild symptoms over a short duration. One parent said, "he just had mild symptoms that slowed him down for really a day or [a day] and a half. And then they disappeared." (Interview 25, parent of an eight-year-old).

One parent said, "my kids were asymptomatic, which was nice. It was a big stress relief. It was a little stressful after they tested positive, just waiting to find out if they were going to develop symptoms or not" (Interview 3, parent of a two-and-a-half-year-old and a five-year-old). Parents also expressed that their children's anxiety about getting COVID-19 also decreased once they had a mild COVID-19 experience:

> "*And then that evening he spiked a fever for a couple hours and by the next day at noon he was totally fine and irritated that he was going to be missing out on life for a week and that was it. And his energy level and everything else kind of came back. It was, yeah, he was not scared of the experience and was just like "well I'm glad that's done and over with".* (Interview 19, parent of a ten-year-old)

**Theme 3: Social stigma and stress.** Many of the parents expressed the social challenges of COVID-19 such as social stigma and stress. Two subthemes emerged: stigma changing over time and social factors of stress. Social stigma related to how the parents initially were reluctant to tell others that their child had COVID-19, but that as the illness spread, parents in later interviews did not feel they were as stigmatized. Stigma did contribute to the stress, however as stigma decreased, parents continued to discuss stress related to child care, managing work when their child needed to be at home, and changes in the family roles and dynamics.

*Subtheme 1*: *Stigma changing over time.* All parents in our study reported on the stigma of COVID-19. However, parents reported varying levels of stigma associated with COVID-19 throughout the pandemic. Near the beginning of the pandemic, parents reported a high degree of stigma with being positive for COVID-19. One parent described not wanting "to be the person to shut down the elementary school" (Interview 2, parent of a two-and-a-half-year-old) and how their "boss had contacted [them] and said, "stop telling people that you have COVID"". Additionally, parents expressed how their children were worried about the stigma of COVID-19 within their friend groups:

*"So, I think our girls were nervous about friends finding out and kind of having that whole stigma. [Daughter 1] has experienced friends that have, like, not invited her to some things now because they're saying, you know, "you've had the COVID so now you could give it to us and I'm worried about my family, my grandparents". (*Interview 1, parent of a 19-year-old (at time of the interview))

Another parent (interview 4, parent of a 15-year-old) described the challenge of hearing from other parents when their child was diagnosed "people were messaging me constantly. Where did you get it–I probably got that question 10 times."

As the pandemic progressed, parents felt that the stigma associated with COVID-19 lessened. As one parent expressed, "I'm not the first person that's had COVID, so I think maybe in the beginning of the pandemic, people were really surprised that people were getting COVID, but I think now so many people have had COVID that it's–people are less shocked. . ." (Interview 9, parent of a two-year-old and a four-year-old). Similarly, another parent said, "at the beginning it seemed like kids would be like, "oh, stay away, stay away" but now they understand that it can happen, right." (Interview 10, parent of a 13-year-old). Other parents felt that COVID-19 became normalized as more people were testing positive. One parent stated, "I think it's a lot better now. A couple of my daughter's friends have actually gotten COVID and so I think it's helped to reduce that stigma and now it's like, "well, we both had COVID so now we can hang out, because we've both had COVID"" (Interview 5, parent of a 14-year-old). Parents reported that as COVID-19 became more normalized, their stress subsided. One parent said, "I'm at the point now where COVID has just become such a part of our life, like, I wasn't worried per se." (Interview 20, parent of a 12-year-old, a 13-year-old and a 16-year-old).

*Subtheme 2*: *Social factors of stress.* The majority of the parents in our study felt they needed to continuously adapt and navigate new external and multi-faceted stressors during COVID-19. Specifically, parents expressed that isolation brought about significant challenges, and one parent said that "the quarantine was probably worse than the COVID". For some parents, the pandemic forced families to change their normal household activities. One parent that required isolation said, my (11-year-old) "he was downstairs and he cried because the dog went upstairs. And I'm, like, "what's this?" "I just wanted to pet the dog". (Interview 24, parent of an 11-year-old and a 13- year-old). In some cases, parents expressed the challenges of having one member of the household testing positive for COVID-19: "the contact tracers called us, and they were like, "oh he should go live somewhere else"" (Interview 17, parent of an 18-month-old).

In addition to logistical challenges of isolation, parents expressed that managing work from home was a source of stress. One parent reported, "I didn't realize how stressed out I was and just the thought of working from home even, like, when I'm trying to manage all the meals and then bring him snacks and then, you know, doing laundry and wearing gloves" (Interview 5, parent of a 14-year-old). On top of navigating their work situation, parents were also dealing with school and childcare stressors. Parents reported having to step into multiple roles, such as becoming teachers, which required their additional time and energy. A mother reported that "And it became just this constant source of fighting where all I did all day was nitpick at him to be on the computer and pay attention and, you know, like it was just, I'm not your mother anymore." (Interview 26, parent of a 16-month-old and a six-year-old).

While these logistical and external stressors were challenging, parents also felt it was difficult to witness the social aspect of their children isolating. One parent said "it was tough. It was a big test in my ability to parent all these teenagers who are so social. And, yeah, it's much nicer now on the other side of it, yeah." (Interview 5, parent of a 14-year-old). For some

parents, these social challenges were experienced by the whole family: "we, as a family we missed out on so many things. And especially for my youngest one, I feel so bad for him not playing soccer and not meeting his friends, and he gets COVID." (Interview 14, parent of a 14-year-old).

Isolation guidelines during COVID-19 also meant that children faced a lack of socialization with their peers. Consequently, parents expressed these social factors significantly impacted their children. One parent said, "My three-year-old. . . she's a little weird around people now, just not as open" (Interview 11, parent of a three-year-old). Similarly, parents reported that these social factors had an impact on their children's attachment to home: "my two-and-a-half-year-old doesn't really leave the house. So, if we go out now he goes "get in the car, I want to go home"" (Interview 3, parent of a five-year-old). Parents reported that these attachments often led to difficult adjustments for their children. One parent expressed, "Going back to school it's a lot more adjustment than I thought it would be" (Interview 16, parent of a six-year-old). Constantly adapting to these multi-faceted logistical and social stressors meant that parents felt burnt out and tired. As one parent stated, "Yeah, I'm pretty worn out. I was pretty excited to send her to daycare today." (Interview 18, parent of a two-year-old).

**Theme 4: A sense of community.** This theme related to parents sharing positive stories about "getting through it" together, to being able to reach out to their community for help. Parents shared how technology enabled them to stay in touch with friends and families, and how their community supported them during isolation.

Despite the challenges that surfaced for families while caring for a child with COVID-19, parents in our study highlighted the importance of community. Parents expressed that it was critical for them to reach out to their community as a source of support. A parent stated that "the sense of community is very alive when you have something like that and you share it with people too, right. If you don't tell them, then they're not going to know. But if you can, you know, tell them, and they can help you." (Interview 12, parent of a six-year-old).

For the majority of parents, they expressed that their family and community were positive sources of support. One parent stated, "there were so many people who were sick and the community was aware, and like tons of people were like whatever you need, I'll drop off groceries" (Interview 23, parent of 2 an eight-year-old and a ten-year-old). Parents expressed they felt support given that family and community members offered to help in any way they could. A parent said, "we have friends that went and did grocery runs, medication runs for us. We had people that dropped off meals for us" (Interview 13, parent of a four-year-old).

During isolation, the sense of community was strengthened when parents were permitted to isolate with their children. One parent expressed, "because we were isolated together, we could still have that physical interaction. We could still have kisses and everything because we were both positive so it was, "ah whatever, doesn't matter"" (Interview 2, parent of a two-and-a-half-year-old).

In some cases, parents reported that technology was integral to feeling supported. One parent said, "we would FaceTime every day because we're all in lockdown–and they were doing puzzles and ordering Skip the Dishes, and having a grand ol' time over there" (Interview 5, parent of a 14-year-old). Another parent highlighted that technology brought people together during this time: "Like our next-door neighbors, we're friends with on Facebook, they dropped off a bag of chocolate bars one night." (Interview 4, parent of a 15-year-old). Additionally, technology allowed more services to be accessible for parents isolating. A parent expressed, "We're also pretty fortunate, like, the time–the era that we live in, because you know, you can order your groceries online and somebody just brings them right to your door" (Interview 9, parent of a two-year-old and a four-year-old).

## Discussion

This study provides insight into the experiences of parents caring for a child with COVID-19. Our findings suggest that parents' experiences are diverse and multifaceted. Specifically, parents' experiences were shaped by the evolving nature of the COVID-19 illness and changing information and guidelines. Further, their experiences evolved and shifted over the course of the pandemic as more people became infected.

Parents in this study highlighted how the stigmatization of COVID-19 impacted their experiences of caring for a child with COVID-19. Specifically, at the start of the pandemic parents described feeling a sense of shame when disclosing theirs or their child's positive COVID-19 test with family, friends, and employers. Feelings of shame heightened when other individuals, such as employers, stigmatized them for sharing their COVID-19 diagnosis. These findings align with a previous qualitative study which found that parents' perceptions of how others would react impacted parents' willingness to get tested or inform others [26]. It is important to note that in our study, the stigma associated with COVID-19 was prominent within the interviews that occurred during the beginning of the pandemic. As the pandemic progressed and the rate of people testing positive for COVID-19 increased, the parents in our study felt that COVID-19 was becoming normalized and part of their daily lives. This finding illuminates that parents likely became more desensitized to the effects of the pandemic whereby COVID-19 and the corresponding protocols became the 'new normal'. While the stress associated with the stigma of COVID-19 lessened across the participants' experiences since the start of the pandemic, the parents in our study identified other sources of stress that are critical to examine such as navigating inconsistent information, uncertainties with COVID-19 symptoms in children, and unrealistic isolation guidelines.

The novelty of the COVID-19 pandemic has produced feelings of uncertainty worldwide [27]. According to Brashers, "Uncertainty exists when details of situations are ambiguous, complex, unpredictable or probabilistic; when information is unavailable or inconsistent; and when people feel insecure in their own state of knowledge or the state of knowledge in general" [28]. Uncertainty-identity theory argues that humans are motivated to reduce feelings of uncertainty by seeking out information through peer interactions and identifying in a group setting [29]. Through our findings, participants sought out or hoped for a strong sense of community to identify with. Uncertainty-identity theory helps illuminate the links between the high degree of uncertainty as a result of social circumstance during the COVID-19 pandemic and parents' experiences and information needs that gravitated toward community-based and group interaction.

The beginning of the pandemic was marked with uncertainty from healthcare providers, policymakers, and leaders given the novelty of the disease. This uncertainty meant that healthcare providers were devoid of clear, evidence-based information to guide the public and their patient care [30]. As Paek and Hove observed uncertainty arose during the pandemic due to divergent views within the cohort of experts [31]. This highlighted the need to improve how to communicate uncertainty to the public without confusion or loss of trust. The impact of this uncertainty was experienced by the participants of our study. Parents in our study highlighted that it was difficult to care for a child with COVID-19 given the changing yet unclear information. The lack of clear, consistent, and up-to-date information and messaging from health officials meant that parents felt uncertain with how to navigate their healthcare decision-making for their child with COVID-19. These findings align with the existing literature describing that the contradictory information left caregivers feeling uncertain about what COVID-19 information to trust [32]. Carney and colleagues found that inconsistencies in health messaging were linked with parents' feelings of frustration [33]. Our findings add to this evidence by

highlighting that these uncertainties were compounded by the lack of pediatric specific information on the presentation and prognosis of COVID-19 in children. Parents particularly described that it was challenging to navigate the 'strangeness' of COVID-19 given the limited information with how COVID-19 impacts the physical health of children. These findings are congruent with Marino et al.'s study that found parental frustration and concerns with the limited pediatric-focused COVID-19 information and guidance to support families during the pandemic [34]. For the parents in our study, not knowing the severity of COVID-19 in children was linked with their anxiety levels.

Parents in our study indicated that their anxiety about their child testing positive for COVID-19 subsided after the illness had run its course and they realized it was for the most part mild symptoms. Given that parents in our study reported asymptomatic or mild symptoms for their child with COVID-19, parents felt a sense of relief once their child had COVID-19. While our study sample does not reflect all COVID-19 cases in children, these findings suggest it is critical to develop clear, consistent, and reliable COVID-19 information that outlines how COVID-19 affects the health of children. The use of clearer COVID-19 messaging may better equip parents when caring for their child with COVID-19 by helping parents navigate the uncertainties of a COVID-19 diagnosis.

While there are several accessible sources to receive COVID-19 information, not all resources are reliable or contain the most up-to-date evidence. Caulfield et al. discussed that this novel disease has evolved at a rapid pace and the scientific community continues to face challenges in communicating research findings that often have conflicting results compared to exaggerated headlines. These factors have the potential to increase confusion and erode public trust [35]. As discussed by Saitz and Schwitzer, poor communication by governments and the scientific community during COVID-19 may jeopardize the public's trust [36]. Our participants reported turning to the internet and media for easily accessible information and to their healthcare providers for additional guidance. This finding aligns with other studies that explore parents' health information needs for acute pediatric illnesses such as pediatric fevers and bronchiolitis [37, 38], and during COVID-19 [26, 39]. However, Arlinghaus, Hersch and Neumark-Sztainer report that turning to popular media and other news sources may contribute to feelings of uncertainty, confusion, and stress for parents [40]. Studies suggest that given the use of the internet and media as a prominent source of information for COVID-19, the use of clear communication when disseminating on social platforms and other media channels is important [26, 39]. Our study supports this need for clear and consistent evidence-based online information as an avenue for parents' effective healthcare decision-making and as a strategy to mitigate parents' anxiety related to COVID-19.

In addition to navigating uncertainties with COVID-19 symptoms and guidelines, parents in our study expressed that the information and guidelines were not always relevant or realistic for their specific context. For example, some parents described their experience of being advised to relocate outside the family home. While this was feasible for some parents, this was not always realistic for other families due to childcare, financial, or other logistical reasons. Similarly, Hodson et al.'s study found that parents' access to available resources such as childcare, size of the home, and local shops for groceries affected the ability for families to effectively isolate according to the public health guidelines [41]. Another study reported that access to resources created barriers for families to follow COVID-19 isolation measures, despite their willingness to adhere to such protocols [26]. Furthermore, all parents in our study expressed the challenges of continuously adapting to external pressures when caring for a child with COVID-19 including managing childcare, working from home, and the lack of socialization. These findings are congruent with previous studies that identify parents' access to resources including extended family or other social support, ability to take time off work, income level,

access to technology, and size of family home as influential factors when caring for a child with COVID-19 [26, 41, 42]. Carney, Behrens and Miller-Graff also highlighted parents' feeling of frustration that community resources exist but are often inaccessible for many [33]. In combination with other literature, our finding highlights that attention to the social determinants of health and specific social contexts are critical considerations when developing COVID-19 information and guidance for parents.

Parents in our study discussed how challenging it was to transition to home schooling and become their child's primary teacher, in addition to working from home themselves. Most of the parents in our study were mothers, and as highlighted by McGrath et al, the effects of homeschooling in addition to traditional gender roles whereby women still bear the burden of household chores and childcare impacted their psychological well-being [43]. COVID-19 stressors have been linked with worse mental health outcomes in women, including increased time demands due to child care and chores, social isolation, occupational stress and maladaptive coping strategies [44].

As highlighted by Chanchlani, Buchanan and Gill, in order to better support families during a pandemic, we require a more comprehensive understanding of the personal and practical context that influence decision making [45]. Understanding the experiences of parents with a child with COVID-19 is therefore critical for healthcare providers and researchers when developing information that is useful for families. The results from this study are essential for developing health information that meets these unique needs of parents caring for a child with COVID-19. These findings will contribute to the development of a knowledge translation tool about caring for a child with COVID-19.

## Limitations

Since our interviews were conducted on Zoom, our sample only included participants with access to technology. We relied on parents' self report of their experiences and therefore recall bias is a potential limitation. Despite extensive efforts to ensure variation in our sample, the majority (84%) of our participants were mothers, who were well educated with high incomes. As a result, our findings may not fully reflect the experiences and information needs across different parent groups.

## Conclusion

Our findings demonstrate that caring for a child who has COVID-19 has significant negative effects on children and their parents. Our findings highlight that parents struggled to deal with the uncertainty of this novel disease including changing rules, difficulty accessing reliable information that was specific to their child's needs, and not knowing how ill their child may become once they tested positive. In addition, they struggled to manage the stress associated with isolation requirements. Our parents did report however that the support they received from their community lessened the burden of caring for their child, and that the stigma of having their child diagnosed with COVID-19 lessened over time as the illness became more widespread. Having access to reliable and trustworthy resources on how to manage COVID-19 and when to seek healthcare services would help mitigate the stress and anxiety parents often experience while caring for a sick child.

## Supporting information

**S1 File. Screening criteria.**
(DOCX)

**S2 File. Demographics questions.**
(DOCX)

**S3 File. Interview guide.**
(DOCX)

## Acknowledgments

We would like to acknowledge the parents and families who shared their experiences with our research team and Una Mehrotra who assisted with administrative aspects of the study.

## Author Contributions

**Conceptualization:** Lisa Hartling, Shannon D. Scott.

**Data curation:** Kathy Reid, Shannon D. Scott.

**Formal analysis:** Samantha Louie-Poon, Priscilla O. Appiah.

**Funding acquisition:** Lisa Hartling.

**Resources:** Shannon D. Scott.

**Supervision:** Shannon D. Scott.

**Writing – original draft:** Samantha Louie-Poon, Kathy Reid.

**Writing – review & editing:** Samantha Louie-Poon, Kathy Reid, Priscilla O. Appiah, Lisa Hartling, Shannon D. Scott.

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
