## [Editor Report · Decision Letter 0]

20 Feb 2023

PONE-D-23-01031“There is a strangeness in this disease”: A qualitative study of parents’ experiences caring for a child diagnosed with COVID-19PLOS ONE

Dear Dr. Louie-Poon,

Thank you for submitting your manuscript to PLOS ONE. After careful consideration, we feel that it has merit but does not fully meet PLOS ONE’s publication criteria as it currently stands. Therefore, we invite you to submit a revised version of the manuscript that addresses the points raised during the review process.

We look forward to receiving your revised manuscript.

Kind regards,

Miwako Hosoda

Academic Editor

PLOS ONE

Journal Requirements:

**Additional Editor Comments:**

I think this is a very significant paper that interviews and carefully analyzes the various difficulties that parents of children affected by COVID-19 feel.　

However, it would be required to show some robust theoretical frameworks such as Uncertainty and Health Literacy, with references provided. The survey included those who have children under the age of 18. Were there differences in conclusions based on the age of the children? Parents' opinions may vary depending on whether they are babies, toddlers, children, or teenagers.

---

## [Author Response · Author response to Decision Letter 0]

29 Mar 2023

Thank you for your insightful comments and suggestions. Please see below for our responses to each of the following comments. 

We have revised our manuscript to meet PLOS ONE’s style requirements. 

We have revised our cover letter to include details on the ethical restrictions for data being available upon request. Please see the second paragraph of our revised cover letter. 

We have updated our reference list in track changes in our revised manuscript submission to reflect the updated references added to our manuscript at this stage of revisions. 

Thank you for this comment and for providing valuable feedback on the use of uncertainty and health literacy theoretical frameworks. We did not set an a-priori theoretical framework to interpret or analyze our findings. While we did not utilize uncertainty and health literacy theory to underpin our philosophical and methodological position from the start, we do believe these frameworks provide useful context to ground our findings. Therefore, we have now expanded our discussion section to include these theories, please see track changes on pages 19, 20, 21. Particularly, Brashers (2001) conceptualization of uncertainty was referenced to introduce uncertainty and provide further context on uncertainty in relation to heath literacy and information (see page 19). Hogg’s (2007) uncertainty-identity theory, originating from psychology, was cited to illuminate the links between the uncertain conditions produced by COVID-19 and parents’ experiences of wanting a sense of community (see page 19-20). On pages 20 and 21, other works were cited to provide uncertainty within the context of COVID-19 and our findings. 

Thank you for this comment. While this is indeed an important question to ask, the purpose of our research question was to explore the experiences and information needs of parents of children diagnosed with COVID-19 throughout the pandemic. Our aim was not to conduct a comparative analysis of children by age or development age, and therefore, understanding differences based on age is not within the purpose or scope of this study.

---

## [Editor Report · Decision Letter 1]

15 May 2023

PONE-D-23-01031R1“There is a strangeness in this disease”: A qualitative study of parents’ experiences caring for a child diagnosed with COVID-19PLOS ONE

Dear Dr. Scott,

Thank you for submitting your manuscript to PLOS ONE. After careful consideration, we feel that it has merit but does not fully meet PLOS ONE’s publication criteria as it currently stands. Therefore, we invite you to submit a revised version of the manuscript that addresses the points raised during the review process.

We look forward to receiving your revised manuscript.

Kind regards,

Melanie Cooper, PhD

Academic Editor

PLOS ONE

Journal Requirements:

Additional Editor Comments:

If you have data on the age of the children, please could you include this in the findings and discussion as the age of the child could have a massive effect on the experience of the parents. An example could be a baby that has developed a severe cough could be more alarming than in a 17 year old. This would not lead to a comparison of findings, it would just provide extra context that influences the parents' experiences and could influence the future development of a knowledge translation tool.

---

## [Author Response · Author response to Decision Letter 1]

22 Jun 2023

Hello, Thank you for the very kind review of our manuscript. We have included a response to reviewer table that itemizes and responds to each reviewer comment. Most sincerely,

---

## [Editor Report · Decision Letter 2]

7 Jul 2023

PONE-D-23-01031R2“There is a strangeness in this disease”: A qualitative study of parents’ experiences caring for a child diagnosed with COVID-19PLOS ONE

Dear Dr. Scott,

Thank you for submitting your manuscript to PLOS ONE. After careful consideration, we feel that it has merit but does not fully meet PLOS ONE’s publication criteria as it currently stands. Therefore, we invite you to submit a revised version of the manuscript that addresses the points raised during the review process.

We look forward to receiving your revised manuscript.

Kind regards,

Melanie Cooper, PhD

Academic Editor

PLOS ONE

Journal Requirements:

**Additional Editor Comments:**

Thank you for including the age of the child in the results section. However, please include a discussion of the influence of the age of the child in the discussion section. Did you notice a difference in for example the findings around anxiety based on the age of the child? Did the age of the child appear to influence any of the other themes and if it did, how does this compare with previous literature? Would you have any different recommendations based on this?

---

## [Author Response · Author response to Decision Letter 2]

29 Aug 2023

Dear Editor, 

Thank you for your insightful comments and suggestions. Please see below for our responses to each of the reviewer/editor comments that have enhanced our submission.

Reviewer Comments Response and revisions made

Thank you for including the age of the child in the results section. However, please include a discussion of the influence of the age of the child in the discussion section. Did you notice a difference in for example the findings around anxiety based on the age of the child? Did the age of the child appear to influence any of the other themes and if it did, how does this compare with previous literature? Would you have any different recommendations based on this?

 Thank you for your comments about the age of the children with COVID-19. In response to this comment we have:

-added Table 2 (lines 146) which delineates the ages of the children with COVID-19

-Lines 493 – we have added that “parents reported anxiety regardless of the age of their child with COVID-19”

-Lines 503 – we have added “There was little difference in parents’ anxiety based on the age of their child however parents’ reported that their anxiety was tied to different events (e.g., online learning). 

-Line 565 – In the limitation section we added that comparing parental experiences based upon the ages of the child with COVID-19 extended beyond the primary focus of this paper.

---

## [Editor Report · Decision Letter 3]

16 Jan 2024

PONE-D-23-01031R3“There is a strangeness in this disease”: A qualitative study of parents’ experiences caring for a child diagnosed with COVID-19PLOS ONE

Dear Dr. Scott,

Thank you for submitting your manuscript to PLOS ONE. After careful consideration, we feel that it has merit but does not fully meet PLOS ONE’s publication criteria as it currently stands. Therefore, we invite you to submit a revised version of the manuscript that addresses the points raised during the review process.

**Please provide a checklist such as **SRQR or COREQ==============================

We look forward to receiving your revised manuscript.

Kind regards,

Karolina Linden, Ph.D

Academic Editor

PLOS ONE

Journal Requirements:

Additional Editor Comments:

Thank you for revising your manuscript. I have just been assigned as the new academic editor to your manuscript. I am sure that it is frustrating with yet another person being involved in the process but please bear with me. I have gone through your submission and the two revisions, and I am happy with the responses that you have provided. However, I cannot seem to locate any checklist such as SRQR or COREQ. I especially miss a discussion about trustworthiness of data and transferability of findings. Please provide such a checklist and expand where needed in the manuscript. I promise to make the process as quick as I can on my side.

Best wishes,

Karolina Linden

---

## [Author Response · Author response to Decision Letter 3]

30 Jan 2024

We have supplied the COREQ checklist. We have attached the response to reviewers addressing the additional discussion regarding trustworthiness of the data and transferability of the findings. We added further references to the methodology section regarding the sample and data collection to reflect this. Please also find attached the updated manuscripts including one with tracked changes and the updated cover letter. Thank you

---

## [Decision Letter · Decision Letter 4]

5 Feb 2024

PONE-D-23-01031R4“There is a strangeness in this disease”: A qualitative study of parents’ experiences caring for a child diagnosed with COVID-19PLOS ONE

Dear Dr. Scott,

Thank you for submitting your manuscript to PLOS ONE. After careful consideration, we feel that it has merit but does not fully meet PLOS ONE’s publication criteria as it currently stands. Therefore, we invite you to submit a revised version of the manuscript that addresses the points raised during the review process.

**Thank you for re-writing your manuscript in accordance with COREQ guidelines. Three minor points to address before the manuscript can be accepted. Please see below. **==============================

We look forward to receiving your revised manuscript.

Kind regards,

Karolina Linden, Ph.D

Academic Editor

PLOS ONE

Journal Requirements:

Additional Editor Comments:

Thank you for preparing your manuscript in accordance with COREQ. I have gone through your manuscript myself as well as secured one external reviewer who had very valid points. There are three minor adaptations left to conduct before the manuscript can be accepted.

1. Please add heading study design to the methods section in accordance with COREQ

2. Illustrate an example of your analysis from code, sub theme to theme in a table.

3. Please add a results table that illustrate Themes and their respective sub themes.

Reviewers' comments:

Reviewer's Responses to Questions

**Comments to the Author**

1. If the authors have adequately addressed your comments raised in a previous round of review and you feel that this manuscript is now acceptable for publication, you may indicate that here to bypass the “Comments to the Author” section, enter your conflict of interest statement in the “Confidential to Editor” section, and submit your "Accept" recommendation.

Reviewer #1: All comments have been addressed

2. Is the manuscript technically sound, and do the data support the conclusions?

Reviewer #1: Yes

3. Has the statistical analysis been performed appropriately and rigorously? 

Reviewer #1: N/A

4. Have the authors made all data underlying the findings in their manuscript fully available?

Reviewer #1: No

5. Is the manuscript presented in an intelligible fashion and written in standard English?

Reviewer #1: Yes

6. Review Comments to the Author

Reviewer #1: Thank you for allowing me to review this important manuscript. The authors have responded well to the previous reviewers comments. I have only a few comments to make the paper adhere to the COREQ guidelines.

Please add a heading Study design.

I would like to see an example of your analysis from code, sub team to team, which can be presented in a table.

The manuscript would also benefit of having a results table that illustrate Teams and their respective sub teams.

7. PLOS authors have the option to publish the peer review history of their article (what does this mean?). If published, this will include your full peer review and any attached files.

Reviewer #1: No

---

## [Author Response · Author response to Decision Letter 4]

20 Feb 2024

Thank you for the review. Please find attached the updated manuscript with the heading Study Design included as well as a table illustrating analysis example from code to subtheme to theme, and a table of themes and subthemes.

---

## [Editor Report · Decision Letter 5]

22 Feb 2024

“There is a strangeness in this disease”: A qualitative study of parents’ experiences caring for a child diagnosed with COVID-19

PONE-D-23-01031R5

Dear Dr. Scott,

We’re pleased to inform you that your manuscript has been judged scientifically suitable for publication and will be formally accepted for publication once it meets all outstanding technical requirements.

Kind regards,

Karolina Linden, Ph.D

Academic Editor

PLOS ONE

Additional Editor Comments (optional):

Thank you for addressing the reviewer comments. I am happy to accept this manuscript for publication.
---

## [Editor Report · Acceptance letter]

22 Mar 2024

PONE-D-23-01031R5 

PLOS ONE

Dear Dr. Scott, 

I'm pleased to inform you that your manuscript has been deemed suitable for publication in PLOS ONE. Congratulations! Your manuscript is now being handed over to our production team.

Kind regards, 

on behalf of

Dr. Karolina Linden 

Academic Editor

PLOS ONE